# In situ molecular weaving of ionic polymers into metal-organic frameworks for radioactive anion capture

Xinghao Li[1], Xiang Lin[1], Zhenzhen Feng[1], Feng Chen[1], Qihang Huang[1], Linlin Zheng[1], Hongwei Wu[2], Jiayin Yuan[3], Yaozu Liao[1] ✉ & Weiyi Zhang[1] ✉

Encapsulation of polymer chains into nanochannels of metal-organic frameworks (MOFs) to construct polymer-MOF hybrid materials can retain the desired properties of two disparate materials. However, the facile fabrication of such hybrids remains challenging, given the difficulty in unraveling entanglement of polymer chains and constraining them into ordered conformations. Herein, we introduce an in situ molecular weaving strategy to construct ionic polymer-MOF hybrid materials, employing shear forces and coordination interactions to untangle cationic polymer chains and guide their realignment within MOF nanochannels during framework formation. Notably, this realignment promotes uniform polymer distribution and exposes more anion-exchange sites. The resulting hybrids capture $ReO_4^-$ (a nonradioactive surrogate of $^{99}TcO_4^-$) with a capacity of 438 mg g$^{-1}$ and reach adsorption equilibrium within 20 min. Characterization and theoretical calculations reveal that the hydrophobic pores in the hybrid materials confer strong affinity toward less hydrated $^{99}TcO_4^-$ anions, thereby enhancing their selectivity over competing anions.

Metal–organic frameworks (MOFs), constructed by organic ligands and metal ions/clusters, have emerged as important porous functional materials[1,2]. Given the high specific surface areas, topological diversity and facile functionalization, MOFs are often used as a host to load guest molecules for separation, catalysis, biosensing, drug delivery diagnostics, etc[3–7]. Recently, in addition to small molecules, large polymer chains have also been introduced as guest substances into MOFs to construct polymer-MOF hybrid materials[8–12]. The marked distinctions between the nature of MOFs and polymers have prompted efforts to utilize the benefits of polymers (e.g., good processability and chemical stability) together with the traits of MOFs (e.g., crystallinity, well-determined structures, and permanent porosity)[13,14]. Conventional methods for fabricating polymer-MOF hybrid materials typically involve introducing monomeric precursors into MOF nanochannels followed by in situ polymerization[15,16]. However, these approaches face critical limitations. Typically, MOFs often require high-temperature activation to remove residual guests and enhance pore accessibility. And during polymerization, thermal insulation within MOF nanochannels reduces pore temperature, diminishing efficiency and yielding poorly defined polymers with low conversion rates. Furthermore, the low polymer loading achieved after the initial polymerization often necessitates multiple cycles to reach the desired level of incorporation. This iterative process complicates synthesis and undermines reproducibility. These challenges highlight an urgent need for innovative strategies that overcome these limitations while retaining the structural and functional benefits of MOFs.

Molecular weaving is a cutting-edge technique for constructing materials with precise molecular architectures, achieved by aligning

[1]State Key Laboratory of Advanced Fiber Materials, College of Materials Science and Engineering, Donghua University, Shanghai 201620, China. [2]College of Chemistry and Chemical Engineering, Donghua University, Shanghai 201620, China. [3]Materials Chemistry Division, Department of Chemistry, Stockholm University, Stockholm 10691, Sweden. ✉e-mail: yzliao@dhu.edu.cn; wyzhang@dhu.edu.cn

molecules through hydrogen bonding, π–π stacking, and host–guest interactions, etc[17–20]. When integrated with crystalline porous frameworks like MOFs, this method utilizes their regular, periodic nanochannels and tunable pore environments to guide the spatial arrangement of substances[21–24]. The periodic structure of MOFs serves as an ideal template, ensuring uniform molecular alignment. By incorporating polymer chains into MOF nanochannels, in situ molecular weaving may occur and emerge as a particularly advantageous approach. Combining such technique with constructing process of MOFs at the same time, it enables the simultaneous synthesis of MOFs and the precise alignment of polymer chains, ensuring structural compatibility and minimizing mismatches. By harnessing the synergy between molecular weaving and the porous architecture of MOFs, in situ molecular weaving provides a robust pathway for fabricating polymer-MOF hybrid materials with superior structural precision and new functionality.

Technetium-99 ($^{99}$Tc), as one of the fission products, is a typical radioactive waste with extreme toxicity, high fission yield (6.06% in the typical uranium fission reactor) and persistent radioactivity (half-life of $2.13 \times 10^5$ years)[25–27]. Obviously, $^{99}$Tc removal from high-level waste steam prior to vitrification is of great essential. The main form of $^{99}$Tc in industry is pertechnetate ($^{99}$TcO$_4^-$), which may potentially leak into the environment and enter the food chain upon accidental release[28]. Considering the anionic nature of $^{99}$TcO$_4^-$, tremendous efforts have been devoted to developing cationic framework materials to scavenge $^{99}$TcO$_4^-$ owing to their strong and specific electrostatic attraction and/or host–guest interaction. Referring to the advantages of MOFs' high porosity, the integration of cationic polymers into neutral MOFs is anticipated to yield a variety of polymer–MOF hybrid materials as high-density anion receptors, featuring readily accessible anion-exchange sites[15,29]. This integration may also endow polymer–MOF hybrid materials with fast sorption kinetics and high stability. However, the straightforward integration of cationic polymers into MOFs suffers from size incompatibility, primarily attributed to the constrained dimensions of the open windows in MOFs. Besides, most cationic polymer chains introduced via this approach tend to entangle and thus disfavor the exposure of active sites. In addition, considering the great number of competing anions, e.g., SO$_4^{2-}$ and NO$_3^-$ in the

natural waste system, it is crucial to design and synthesize highly effective sorbents for $^{99}$TcO$_4^-$.

Herein, we propose a general and mild synthetic approach inspired by in situ molecular weaving to construct ionic polymer–MOF hybrid materials. At first, linear cationic polymers were complexed with anionic coordination organic ligands via electrostatic interactions. During MOF formation, the shear force generated by mechanical stirring, coupled with the directional aligning effect of coordination bonds, acted synergistically to disentangle and orderly align cationic polymer chains within the MOF nanochannels (Fig. 1). This dual-force strategy not only facilitates molecular weaving but also ensures a uniform and stable distribution of cationic polymer chains, overcoming the limitations of in situ monomer introduction followed by polymerization. The resultant hybrid materials exhibit densely packed anion-exchange sites, hierarchical porosity and hydrophobic MOF channels, enabling rapid anion transport and exceptional selectivity for less hydrated anions like $^{99}$TcO$_4^-$. This innovative approach demonstrates the effectiveness of molecular weaving as an effective strategy for constructing polymer–MOF hybrid materials, enabling advanced designs tailored for efficient and selective $^{99}$Tc removal in environmental remediation.

## Results

### Materials synthesis and characterization

The targeted molecularly woven ionic polymer-MOF hybrid materials were synthesized through electrostatic interactions between cationic polymer chains and organic coordination ligands, followed by stirring in a Cu$^{2+}$ solution to reorganize the cationic polymer chains during MOF formation. To start, a cationic polymer, poly(4-cyanomethyl-1-vinyl-1,2,4-triazolium bis(trifluoromethane sulfonyl)imide), was synthesized as a representative form of Ptriaz, according to our previous report[30,31]. In a typical run, carboxylic coordination ligands (benzenetricarboxylic acid, H$_3$BTC) and Ptriaz (the triazolium/-COOH molar ratio = 1:1) were dissolved in DMF to form a homogeneous and transparent precursor solution. In this solution, only negligible complexation occurred, as the -COOH units of carboxylic coordination ligands stayed in a non-dissociated form[32]. Next, the precursor solution was added dropwise into a complexation solution at pH 9–10 adjusted by triethylamine in ethanol, under vigorously stirring followed by

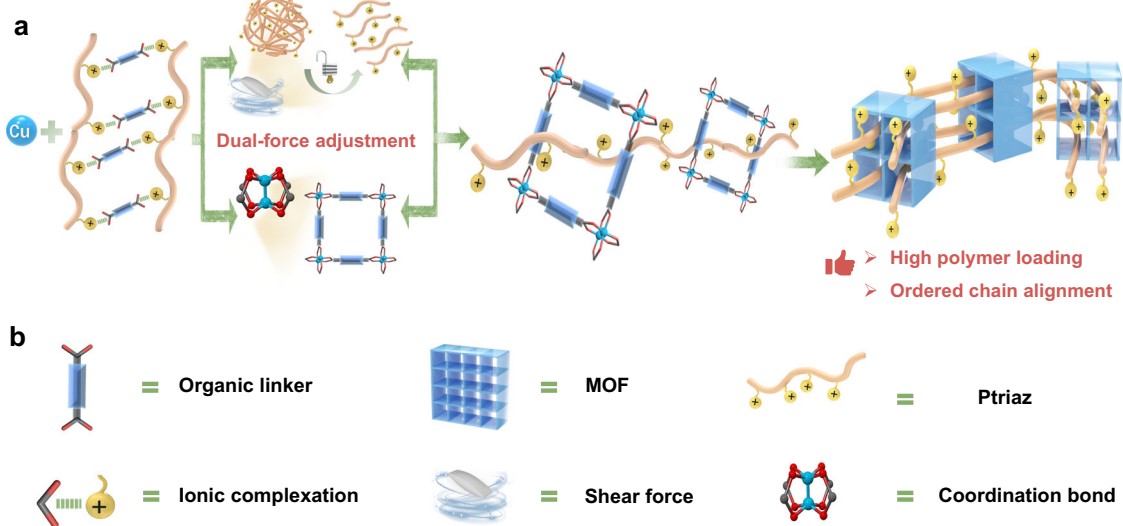

**In situ molecular weaving driven by dual forces**

**Dual-force adjustment**

➤ **High polymer loading**
➤ **Ordered chain alignment**

= **Organic linker**          = **MOF**          = **Ptriaz**

= **Ionic complexation**      = **Shear force**  = **Coordination bond**

**Fig. 1 | Schematic representation of a strategy for synthesizing ionic polymer-MOF hybrid materials. a** Ordered interpenetration of cationic polymer chains within MOF nanochannels, driven by dual-force regulation (coordination bond and shear force) through in situ molecular weaving. **b** Representative symbols for MOFs, ionic monomers, cationic polymers, and interacting forces.

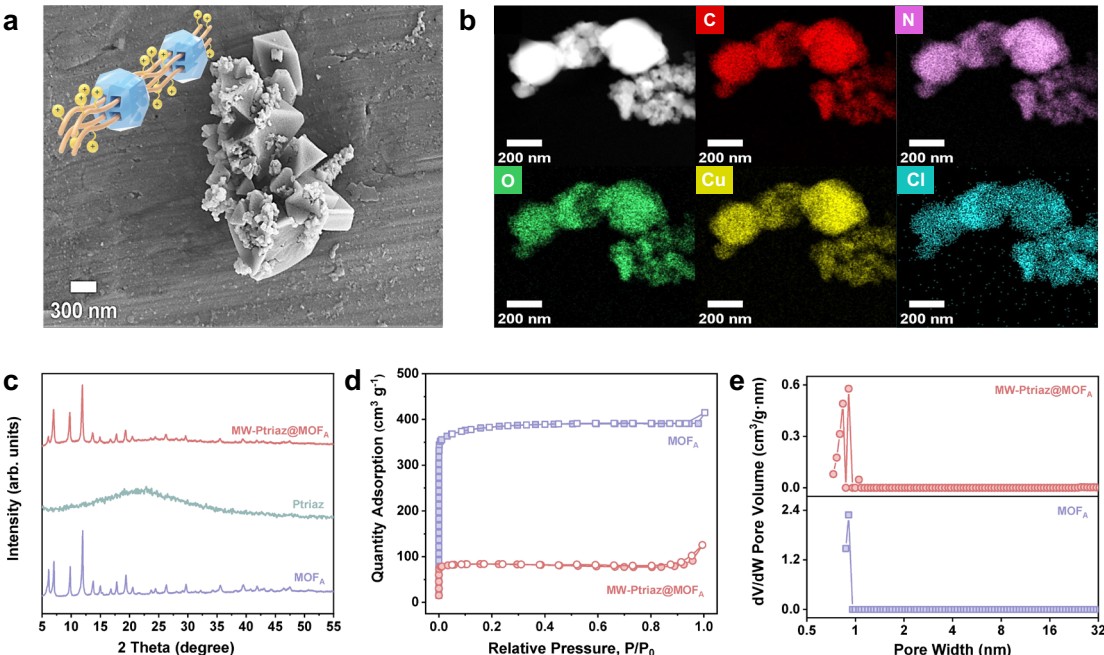

**Fig. 2 | The characterization of MW-Ptriaz@MOF$_A$. a** A graphical representation and an SEM image of the molecularly woven polymer-MOF hybrid material synthesized using MOF$_A$ as a template. **b** STEM images and elemental mapping of MW-Ptriaz@MOF$_A$. **c** PXRD patterns for MOF$_A$, Ptriaz and MW-Ptriaz@MOF$_A$, respectively. **d** N$_2$ sorption and **e** pore size distribution analyses based on NLDFT method for MOF$_A$ and MW-Ptriaz@MOF$_A$, respectively. Source data are provided as a Source Data file.

ultrasonication. The electrostatically complexed product was insoluble and precipitated immediately. At this stage, the deprotonation of -COOH groups in the weakly alkaline environment induced a charged state in the carboxylic coordination ligands (BTC$^{3-}$); it initiated the in situ ionic complexation between the deprotonated carboxylic coordination ligands and the cationic polymer chains to build up the insoluble polymer complexes (IPC). The relevant characterizations of IPC were presented in Figs. S1–S5. The results indicated that IPC exhibited a distinctive aggregated granular morphology, accompanied by an amorphous structure and a relatively low surface area. In the subsequent synthesis steps of molecularly woven ionic polymer-MOF hybrid materials, CuBTC (termed as MOF$_A$) was first selected as the target MOF phase, and the resulting sample was designated as MW-Ptriaz@MOF$_A$. The synthesis involved adding IPC-A into a copper(II) chloride solution prepared in a DMF/EtOH/H$_2$O mixture with a volume ratio of 1:1:1.

Specifically, the mixture was sonicated to disperse the IPC-A followed by annealing at 80 °C under vigorous stirring. The solid was then filtered off and subsequently soaked in EtOH to remove residue reagents before it was dried to form porous materials. It is worth noting that since the coordination interaction between the carboxylate and the Cu$^{2+}$ is stronger than that with the cationic 1,2,4-triazolium cation ring, plus the directionality of coordination bonds, the force generated during the coordination process together with the shear force generated by mechanical stirring further promoted in situ molecular weaving and realign the cationic polymer chains within the ordered porous channels, so that the polymer chains will be evenly and orderly aligned in the nanochannels of MOFs, resulting in the formation of highly structured molecularly woven ionic polymer-MOF hybrid materials. To explore the morphology and size, scanning electron microscopy (SEM) was firstly employed. As shown in Fig. 2a and Fig. S6, compared to MOF$_A$ in an orthododecahedral shape, SEM images of MW-Ptriaz@MOF$_A$ suggested the presence of a depressed pyramid-like morphology of ~100 to ~300 nm in size. In general, the morphologies of the molecularly woven ionic polymer-MOF hybrid materials were not angular and regular as that of MOFs. As such, MW-

Ptriaz@MOF$_A$ synthesized by this approach could generate crystalloid morphologies. The dark-field scanning transmission electron microscopy (STEM) image and its corresponding energy-dispersive X-ray spectroscopy (EDS) elemental mapping of the MW-Ptriaz@MOF$_A$ were shown in Fig. 2b. We observed that the characteristic elements of cationic polymer chains, i.e., nitrogen (N) and chlorine (Cl), are evenly distributed throughout the samples. In particular, the element Cl was originated from the counter ion of the cationic polymers. During the conversion of IPC-A to molecularly woven ionic polymer-MOF hybrid materials, the carboxylic coordination ligands were coordinated with Cu$^{2+}$ in solution, rebalancing electrostatic forces between cationic polymers and the Cl$^-$ (counterion of the copper salt) in solution. However, without stirring, the electrostatic complexation reacted and precipitated in a Cu$^{2+}$ solution can only end up in randomly distributed MOFs particles on the outer surfaces/in the inner pores of the entangled porous cationic polymers (Fig. S7). This phenomenon is related to the lack of shearing force induced by mechanical stirring, which can adjust the orientation of the cationic polymer chains. To be more specific, under the non-stirring condition, the electrostatic complexation between carboxylic coordination ligands and cationic polymers cannot be readily dissociated, forbidding reconstruction of Cu$^{2+}$ with the organic ligands. Consequently, the orderly integration of cationic polymer chains into the MOF nanochannels could not take place as the subsequent step. These results were supported by SEM-EDS elemental mapping spectra (Fig. S8), in which, as observed in non-stirring situation (NS-Ptriaz@MOF$_A$), the characteristic elements of cationic polymer chains, i.e., N and Cl, did not exist in the formed MOF particles. Instead, most MOFs particles were simply grown on aggregated polymers. Powder X-ray diffraction (PXRD) patterns of MW-Ptriaz@MOF$_A$ showed characteristic sharp peaks indexed to MOF$_A$; it demonstrated crystallinity retention of MOF after threading the cationic polymer chains through MOF nanochannels (Fig. 2c). To investigate the pore characters of MOF and molecularly woven ionic polymer-MOF hybrid materials, N$_2$ sorption was investigated at 77 K. The Brunauer-Emmett-Teller (BET) surface areas of 247 m$^2$/g for MW-Ptriaz@MOF$_A$ was diminished compared to that of MOF$_A$ (1115 m$^2$/g)

(Fig. 2d). Compared with MOF$_A$, the pore size distribution of MW-Ptriaz@MOF$_A$ showed the shrinkage of pore sizes based on the non-local density-functional theory (NLDFT) calculation, indicating that the nanochannels of MOF$_A$ were occupied by cationic polymer chains (Fig. 2e). Obviously, the hierarchical pores in molecularly woven ionic polymer-MOF hybrid materials would facilitate the mass transport, and the reduced pore sizes might benefit $^{99}TcO_4^-$ capture. To further investigate whether the dual-force-induced molecular weaving during MOF formation can direct the alignment of cationic polymer chains within the nanochannels of MOFs, rather than leaving them entangled externally, high-speed magic-angle spinning (MAS) $^1H$ solid-state NMR and differential scanning calorimetry (DSC) measurements were performed on MW-Ptriaz@MOF$_A$ and NS-Ptriaz@MOF$_A$, respectively. MAS $^1H$ solid-state NMR analysis (Fig. S9) revealed a characteristic peak at a chemical shift of 7.59 ppm, corresponding to the triazolium hydrogen in Ptriaz. Following in situ molecular weaving, this peak shifted by 2.23 ppm, indicating notable changes in the chemical environment. In contrast, for the sample prepared without stirring (under a single force), the peak shifted by 0.72 ppm. These results demonstrate that the dual forces applied during in situ molecular weaving enhance intramolecular interactions within the pore environment (Fig. S10), leading to improved confinement and alignment of polymer chains. Furthermore, DSC analysis revealed that, unlike NS-Ptriaz@MOF$_A$, MW-Ptriaz@MOF$_A$ exhibited a complete disappearance of the melting peaks associated with Ptriaz (Fig. S11). This observation provided further evidence that the cationic polymer chains were predominantly confined within the MOF nanochannels rather than existing as a simple mixture with the MOFs. Because the melting of Ptriaz was inhibited due to the strong intramolecular interaction between cationic polymer chains and frameworks nanochannels[33,34].

The characterization data of molecularly woven ionic polymer-MOF hybrid materials demonstrated the possible penetration of cationic polymer chains through MOF nanochannels, which matched precisely with corresponding PXRD, BET, MAS $^1H$ solid-state NMR and DSC data. To elucidate the mechanism behind, the radial distribution functions (RDFs) of molecular dynamic (MD) simulations were conducted to provide a molecular explanation for the realignment of cationic polymer chains. RDFs represent the distribution probability of other particles along the distance from a specified particle center, the ordered distribution of different particles in space can be studied and reveal the strength and weakness of interactions between particles[35,36]. To investigate the intramolecular interactions that favor the penetration of cationic polymer chains into the nanochannels of MOFs and their ordered orientation, we computed RDFs for MW-Ptriaz@MOF$_A$. Initially, we established the MOF$_A$ model, incorporating cationic polymer chains to facilitate their penetration into the framework of MOF$_A$ (termed as MOF$_A$-Ptriaz). Subsequently, a detailed examination was undertaken to elucidate the orientation of these polymer chains. As a comparative benchmark, the electrostatic complexes formed between Ptriaz and BTC$^{3-}$ (in the absence of an MOF$_A$ system) were denoted as Ptriaz-BTC. As depicted in Fig. 3a, the RDFs of both models exhibited their first peak at 0.35 nm. In the Ptriaz-BTC system, the RDF curve displayed a much weaker peak than MOF$_A$-Ptriaz, indicating a poor orientation of the Ptriaz with the BTC$^{3-}$ when the interaction was primarily electrostatic. Moreover, for the MOF$_A$-Ptriaz system, the RDFs peak increased significantly upon the coordination of BTC$^{3-}$ with copper ions. This implied an augmented number of cyano groups within equivalent distances, suggesting that MOFs' nanoconfinement for Ptriaz played a crucial role in adjusting the orientation of the Ptriaz, resulting in a more uniform distribution of cyano groups within the Ptriaz. To further analyze the orientation of simulated cationic polymer chains, we specifically identified the C and N atoms in linear Ptriaz and subsequently scrutinized the orientation distribution by examining the dihedral angles in the adjacent structural units of the Ptriaz (Fig. 3b). For Ptriaz, the dihedral angles between adjacent units of C

and N atoms were distributed relatively randomly within the range of 0–180°, and no distinct characteristic peaks were observed, indicating the disorderliness of the cationic polymer chains. When the Ptriaz was confined and systematically arranged within the nanochannels of MOFs, the dihedral angles of Ptriaz predominantly concentrated at 60° and 125°, indicating a substantial increase in the regularity of Ptriaz. Considering that the sum of the angles 60° and 125° is close to 180°, they can be regarded as approximately complementary. Trajectory configuration analysis revealed that the dihedral angle of Ptriaz is 60° when the repeating units are distributed on the same side of the main chain, while it is 125° when the repeating units are distributed on the opposite sides of the main chain. Due to the 90° symmetry in the dihedral angle distribution, this indicates a parallel and uniform distribution of repeating units of Ptriaz on both sides of the main chain, ensuring the stability and ordered orientation of the linear Ptriaz polymer.

Based on the promising results obtained with MW-Ptriaz@MOF$_A$, we further explored the versatility of this approach by expanding the molecularly woven ionic polymer-MOF hybrid materials series. Specifically, we selected CuBDC (termed as MOF$_B$) and CuTCPP (termed as MOF$_C$) as alternative MOF frameworks. Following the same preparation strategy, we successfully synthesized MW-Ptriaz@MOF$_B$ and MW-Ptriaz@MOF$_C$. Comprehensive characterizations, including SEM, STEM-EDS elemental mapping, PXRD, and N$_2$ sorption measurements, confirmed that MW-Ptriaz@MOF$_B$ and MW-Ptriaz@MOF$_C$ exhibited consistent results with those observed for MW-Ptriaz@MOF$_A$. SEM images revealed similar morphological transformations, indicating the formation of hybrid materials with distinct crystalline structures. As shown in Figs. S12–S15, MW-Ptriaz@MOF$_B$ showed a more typical irregular sheet-like morphology of ~150 nm in lateral size, like MOF$_B$. Note that the nanosheets in MOF$_B$ tend to aggregated and stack in squares. Compared to MOF$_C$, which exhibited a stacked quasi-disc morphology with a size of ~1 μm, MW-Ptriaz@MOF$_C$ displayed a morphology resembling nano-flower clusters. STEM-EDS elemental mapping confirmed the uniform distribution of nitrogen and chlorine throughout the samples, further demonstrating the integration of cationic polymer chains within the MOF frameworks (Figs. S16, S17). As observed under non-stirring conditions (NS-Ptriaz@MOF$_B$ & NS-Ptriaz@MOF$_C$), the results were consistent with those of NS-Ptriaz@MOF$_A$, where the majority of cationic polymer chains were not incorporated into the MOF nanochannels. Instead, most MOF particles simply nucleated and grew on the aggregated polymer matrix (Figs. S18, S19). PXRD patterns showed that the crystalline structures of MW-Ptriaz@MOF$_B$ and MW-Ptriaz@MOF$_C$ were retained after threading the cationic polymer chains through their nanochannels (Figs. S20, S21). Furthermore, N$_2$ sorption isotherms revealed a significant decrease in surface areas and pore volume compared to pristine MOF$_B$ and MOF$_C$, indicative of nanochannel occupation by the polymer chains (Figs. S22, S23). Also, MAS $^1H$ solid-state NMR and DSC analyses verified the successful confinement of cationic polymer chains within the MOF nanochannels and their intramolecular interactions with the MOF pores (Figs. S24, S25). For both MW-Ptriaz@MOF$_B$ and MW-Ptriaz@MOF$_C$, MAS $^1H$ solid-state NMR spectra exhibited more evident characteristic shifts in the chemical environment of the triazolium hydrogen, signifying strengthened intramolecular interactions under dual-force-induced molecular weaving (Figs. S26, S27). Similarly, DSC measurements revealed the complete disappearance of the melting peaks associated with Ptriaz, further validating the strong confinement and integration of polymer chains within the MOF nanochannels (Figs. S28, S29).

To benchmark the advantages of our in situ molecular weaving strategy, we systematically compared it with two conventional approaches—in situ polymerization and externally assisted infiltration—in terms of polymer loading efficiency and synthesis conditions (Figs. S30–S32). As summarized in Table S1, molecular weaving

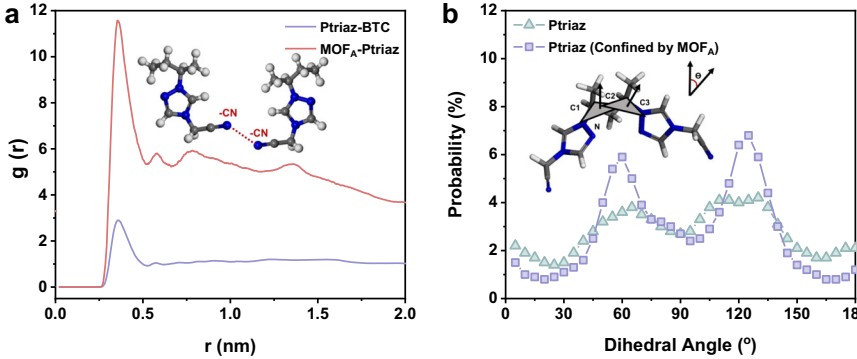

**Fig. 3 | The MD simulations of MW-Ptriaz@MOF_A.** **a** The RDFs of cyano groups in Ptriaz within the cutoff distance in different models. **b** The dihedral angle distribution of Ptriaz and Ptriaz confined within the nanochannels of MOFs, respectively. Source data are provided as a Source Data file.

consistently achieved the highest polymer loading across all MOF systems tested, which is attributed to the dual-force-driven alignment of polymer chains during MOF formation. In contrast, in situ polymerization was hindered by steric effects and incomplete monomer conversion, while externally assisted infiltration was limited by the poor diffusivity of entangled polymer chains. In addition, both conventional methods required high-temperature MOF activation (120 °C, 12 h) and prolonged post-treatment, leading to greater energy consumption and time cost. By eliminating the need for pre-activation and enabling polymer incorporation under more efficient conditions, molecular weaving demonstrates clear advantages in both synthesis efficiency and scalability.

## ReO_4^- scavenging studies

Considering the need for long-term stability of adsorptive materials in the aqueous phase, we selected MW-Ptriaz@MOF_C as the porous adsorbent, which was derived from CuTCPP–MOF featuring hydrophobic porphyrin linkers. Given the predesigned architecture featured by ordered cationic polymer chains throughout MOF nanochannels, it was attempted to employ this water-stable MW-Ptriaz@MOF_C to remove radioactive $^{99}TcO_4^-$ (Fig. 4a). Owing to the limited availability of $^{99}TcO_4^-$ in common laboratories, $ReO_4^-$ was chosen as a non-radioactive surrogate for $^{99}TcO_4^-$ due to their similarity in both magnitude and identical charge density[37]. As shown in Fig. 4b, the adsorption behavior of MW-Ptriaz@MOF_C, MOF_C and Ptriaz was assessed from adsorption isotherms with various initial $ReO_4^-$ concentrations (from 100 ppm to 2500 ppm). According to the adsorption experiments, the functional relationship between adsorption capacity and equilibrium concentration was fitted using Langmuir model and Freundlich model, respectively (Figs. S33–S35). The fitting results indicated that the adsorption of $ReO_4^-$ by three materials (MW-Ptriaz@MOF_C, Ptriaz and MOF_C) all followed Langmuir isotherm model, with high linear correlation coefficient of 0.998, 0.997 and 0.925, respectively. During the comparison of adsorption capacity between cationic polymer (Ptriaz), MOF_C and molecularly woven ionic polymer-MOF hybrid materials (MW-Ptriaz@MOF_C), it was surprisingly found that the MW-Ptriaz@MOF_C had a maximum adsorption capacity of 438 mg/g toward $ReO_4^-$, much higher than that of Ptriaz (188 mg/g) and MOF_C (66 mg/g). The enhanced removal efficiency for $ReO_4^-$ may be attributed to the substantially increasing of exposed cationic adsorption sites, which was originated from realigned cationic polymer chains adjusted by MOFs' ordered nanostructures.

The removal efficiency of $ReO_4^-$ was then investigated by evaluating adsorption kinetics of adsorbents. The experiments were conducted by adding 50 mg adsorbents into 50 mL $ReO_4^-$ aqueous solution (25 ppm). Following mixtures at different time intervals were collecting and separated for inductively coupled plasma optical emission spectrometry (ICP-OES) analysis. As shown in Fig. 4c, for MW-

Ptriaz@MOF_C, the removal efficiency of $ReO_4^-$ increased sharply to over 92% in the initial 1 min. The sorption equilibrium was reached within 20 min (95%), indicating a considerably faster adsorption process than Ptriaz and MOF_C, whose removal efficiency of $ReO_4^-$ reached 83% and 5% within 6 h, respectively. Moreover, this prominent adsorption kinetics can be attributed to the high surface area and hierarchical porous structure of MW-Ptriaz@MOF_C that favors $ReO_4^-$ transfer. In addition, the abundance of precisely tailored 1,2,4-triazolium functional sites in the cationic polymer chains throughout the ordered nanochannels further promoted the capture of $ReO_4^-$. In addition, pseudo-first-order and pseudo-second-order models were applied to fit the adsorption kinetics data of MW-Ptriaz@MOF_C, Ptriaz and MOF_C (Figs. S36–S38). The results indicated that the adsorption kinetics of these three adsorbents towards $ReO_4^-$ can be well fitted with a pseudo-second-order model and the high correlation coefficient indicated that the rate-determining step of the adsorption process between adsorbents and $ReO_4^-$ would be chemical adsorption[38]. Briefly, it is worth mentioning that such fast kinetics process will be beneficial for dealing with the leaked nuclear waste water.

Next, the stability and reusability of adsorbents in harsh chemical environments (alkaline or acidic) are crucial for the practical applications. As shown in Fig. 4d, the PXRD peaks of MW-Ptriaz@MOF_C remained nearly unchanged after being soaked in acidic (pH = 1) or alkaline (pH = 11) water solution, which indicated that the introduction of cationic polymer chains into MOFs nanochannels enhanced the stability of MOF_C in aqueous solutions. After the adsorption of $ReO_4^-$, the PXRD pattern (Fig. S39) showed a broad peak in the 2θ range of 15–35 degrees, which was attributed to electrostatic repulsion between the adsorbed $ReO_4^-$ anions. This repulsive interaction induced lattice alterations in MW-Ptriaz@MOF_C, resulting in a reduction in crystallinity. However, upon desorption of $ReO_4^-$, the regenerated MW-Ptriaz@MOF_C maintained good crystallinity with no significant variations in morphology or microstructure (Figs. S40, S41), further corroborating its structural integrity. The impact of pH on the removal efficiency of $ReO_4^-$ by MW-Ptriaz@MOF_C was also studied, with pH values ranging from 1 to 11. As shown in Fig. 4e, the removal efficiency of $ReO_4^-$ remained consistently high (>80%) across a broad pH range of 3–7. Moreover, as one of the most essential properties of adsorbents in industry, the recyclability of MW-Ptriaz@MOF_C in different aqueous environment was also evaluated. As shown in Fig. 4f and Figs. S42, S43, after 6 cycles of the regeneration process with 2 M NaNO_3 aqueous solution, the removal efficiency of $ReO_4^-$ remained >90% in neutral pH environment. Furthermore, in an acidic environment (pH = 1), the removal efficiency of $ReO_4^-$ by MW-Ptriaz@MOF_C could still maintain 45% after 6 cycles. But in an alkaline environment (pH = 11), MW-Ptriaz@MOF_C exhibited negligible adsorption efficiency for $ReO_4^-$ starting from the third cycle. The observed phenomenon can be attributed to the following factors: at pH = 1, the removal efficiency

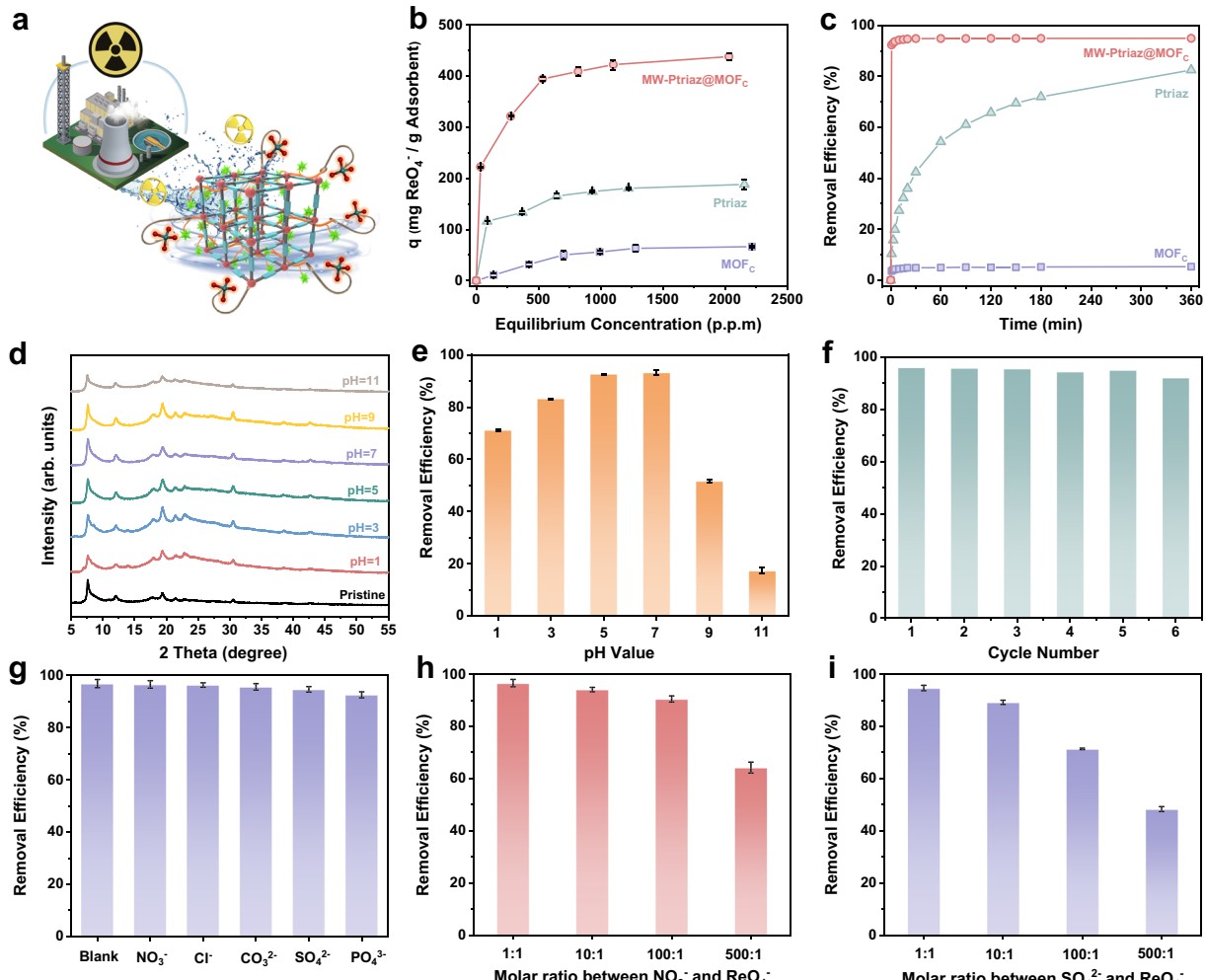

**Fig. 4 | Adsorption performance of MW-Ptriaz@MOF$_C$. a** Illustration of the design of MW-Ptriaz@MOF$_C$ for the efficient sequestration of $^{99}$TcO$_4^-$ from nuclear waste solutions. **b** Adsorption isotherm of MW-Ptriaz@MOF$_C$ (red circle), Ptriaz (green triangle), and MOF$_C$ (purple square) for ReO$_4^-$ uptake, respectively. **c** Adsorption kinetics for MW-Ptriaz@MOF$_C$ (red circle), Ptriaz (green triangle), and MOF$_C$ (purple square), respectively. **d** PXRD patterns of MW-Ptriaz@MOF$_C$ after immersing in the water solutions with pH values varied from 1 to 11. **e** Effect of pH on the ReO$_4^-$ adsorption performances of MW-Ptriaz@MOF$_C$. **f** Reusability of MW-Ptriaz@MOF$_C$ for ReO$_4^-$ removal under neutral pH environment. **g** Effect of typical competing anions on the removal efficiency of ReO$_4^-$. **h** Effect of excess competing NO$_3^-$ anions on the removal efficiency of ReO$_4^-$. **i** Effect of excess competing SO$_4^{2-}$ anions on the removal efficiency of ReO$_4^-$. Error bars represent standard deviation. $n = 3$ independent experiments. Source data are provided as a Source Data file.

relatively decreased due to the competitive interaction between excess NO$_3^-$ and ReO$_4^-$ in the solution. In contrast, in strongly alkaline environments (pH = 11), the excess OH$^-$ anions triggered a ring-opening reaction in the cationic quaternary ammonium groups of 1,2,4-triazolium[26,39], which disrupted the electrostatic attraction between the deionized polymer chains and the negatively charged ReO$_4^-$, leading to a significantly reduced removal efficiency at pH 11. Additionally, PXRD measurements revealed that the MW-Ptriaz@MOF$_C$ maintained good crystallinity after recycling tests in both neutral and acidic environments. In contrast, when subjected to recycling tests in an alkaline environment, the material's crystallinity was significantly compromised (Figs. S44–S46). Above all, the experimental findings demonstrated that MW-Ptriaz@MOF$_C$ has significant potential for effectively removing $^{99}$TcO$_4^-$/ReO$_4^-$ from acidic nuclear waste streams.

For authentic radioactive low activity waste (LAW), there always exist a great number of competing anions, such as NO$_3^-$, SO$_4^{2-}$, Cl$^-$, etc., particularly for NO$_3^-$ and SO$_4^{2-}$ (both coexists in an excess amount of 100 to 1000 folds). Therefore, selectivity of adsorbents was systematically investigated for ReO$_4^-$ capture. Firstly, the removal efficiencies of ReO$_4^-$ by MW-Ptriaz@MOF$_C$ was studied in the presence of NO$_3^-$, SO$_4^{2-}$, Cl$^-$, PO$_4^{3-}$ and CO$_3^{2-}$. As shown in Fig. 4g, when the molar ratios of competing anions were set as identical as that of ReO$_4^-$ in the aqueous solution, the removal efficiencies of ReO$_4^-$ remained >92% in all experiments, revealing a stronger affinity of the adsorbent towards ReO$_4^-$. Additionally, ReO$_4^-$ removal efficiency with the existence of higher concentrated competing anions (NO$_3^-$ and SO$_4^{2-}$) were further evaluated to examine the performance of adsorbents. When the molar ratios of NO$_3^-$ to ReO$_4^-$ were 10:1 and 100:1, the removal efficiencies of ReO$_4^-$ by MW-Ptriaz@MOF$_C$ remained as high as 94% and 91%, respectively. Even the molar ratio of NO$_3^-$ was set as 500-fold in excess, removal efficiency of ReO$_4^-$ could still reach 64% (Fig. 4h). Besides, the removal efficiency of ReO$_4^-$ can still be 48% under an anion molar ratio of SO$_4^{2-}$/ReO$_4^-$ = 500 (Fig. 4i). Such favorable results stem from the hydrophobic backbones of MOF$_C$ composed of porphyrin ligands, which can effectively reduce the affinity between pore walls and strongly hydrated SO$_4^{2-}$/NO$_3^-$ anions, thus enhanced the selectivity for less hydrated $^{99}$TcO$_4^-$/ReO$_4^-$ anions[28]. Following, encouraged by the remarkable performance of MW-Ptriaz@MOF$_C$, we were motivated to investigate the practical applicability of this adsorbent for capturing ReO$_4^-$ in simulated Hanford LAW melter recycle stream. In the Hanford LAW stream (Table S2), the content of Cl$^-$, NO$_2^-$, and NO$_3^-$ were at least 300 times higher than that of $^{99}$TcO$_4^-$, which posed a big challenge for

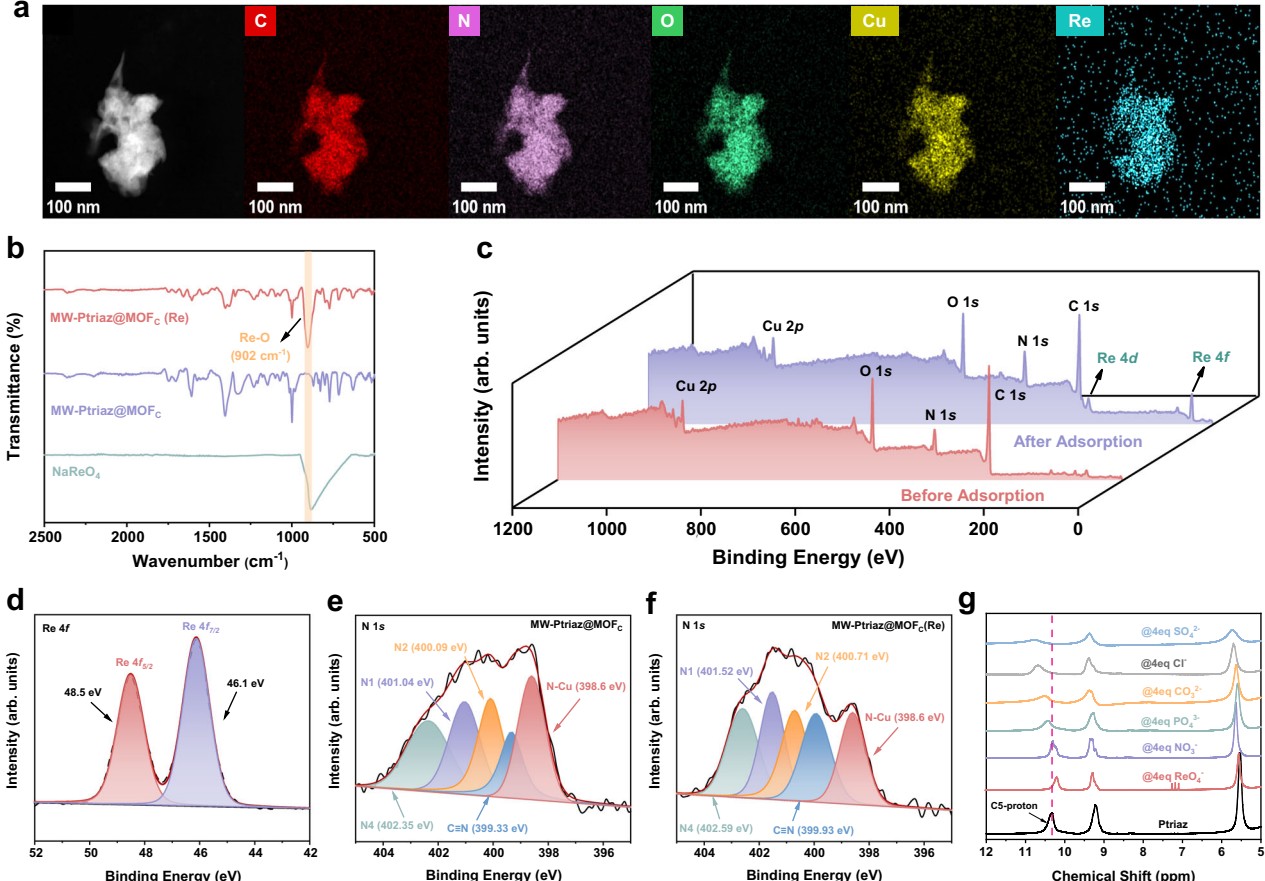

**Fig. 5 | Adsorption mechanism investigation. a** STEM images and elemental mapping for MW-Ptriaz@MOF$_C$(Re). **b** FT-IR spectra of MW-Ptriaz@MOF$_C$ before and after anion-exchange with ReO$_4^-$ and NaReO$_4$. **c** XPS survey of MW-Ptriaz@MOF$_C$ and MW-Ptriaz@MOF$_C$(Re). **d** XPS analysis of Re 4$f$ toward ReO$_4^-$ adsorbed MW-Ptriaz@MOF$_C$. **e** XPS analysis of N 1$s$ survey of MW-Ptriaz@MOF$_C$. **f** XPS analysis of N 1$s$ from ReO$_4^-$ adsorbed MW-Ptriaz@MOF$_C$. **g** $^1$H-NMR spectra of Ptriaz and Ptriaz@4eq anions. Source data are provided as a Source Data file.

its effective capture[37]. Encouragingly, by adding MW-Ptriaz@MOF$_C$ into the simulated nuclear waste solution at a solid/liquid phase ratio of 10:1, the removal efficiency of the MW-Ptriaz@MOF$_C$ to $^{99}$TcO$_4^-$/ReO$_4^-$ can reach up to 66%, indicating its potential in practical applications for authentic scenarios (Table S3).

**Adsorption mechanism**

Zeta potential analysis was performed to investigate the surface charge properties of ionic polymer-MOF hybrid materials. As shown in Fig. S47, the three samples all displayed positive values in aqueous solutions, implying that molecularly woven ionic polymer-MOF hybrid materials with positive charges can effectively immobilize $^{99}$TcO$_4^-$/ReO$_4^-$ anions by electrostatic interactions. The adsorption mechanism was then studied by STEM-EDS, FT-IR spectra and X-ray photoelectron spectroscopy (XPS) analysis. As shown in the STEM-EDS elemental mapping images of MW-Ptriaz@MOF$_C$ after ReO$_4^-$ adsorption (Fig. 5a), it was obvious that the Re element increased significantly in MW-Ptriaz@MOF$_C$(Re). Comparing with the FT-IR spectrum of MW-Ptriaz@MOF$_C$, the new peak at 902 cm$^{-1}$ for MW-Ptriaz@MOF$_C$(Re) can be attributed to Re-O $\nu3$ asymmetric stretching (Fig. 5b)[40], indicating the presence of ReO$_4^-$ after adsorption. The XPS analysis of pristine and Re-loaded MW-Ptriaz@MOF$_C$ was shown in Fig. 5c. Signals of C 1$s$, O 1$s$, N 1$s$, Cu 2$p$ and newly appeared Re 4$f$/Re 4$d$ peaks (at 46 eV/265 eV) could be clearly observed from the wide-range XPS spectra of MW-Ptriaz@MOF$_C$(Re)[26]. As shown in Fig. 5d, the Re 4$f$ core-level high-resolution spectra exhibited two oxidation states of Re, i.e., Re 4$f_{5/2}$

and Re 4$f_{7/2}$, respectively, after adsorption of ReO$_4^-$ in aqueous solutions. The peaks assigned to Re 4$f$ were divided into two peaks corresponding to Re 4$f_{7/2}$ at 46.1 eV and Re 4$f_{5/2}$ at 48.5 eV, which exhibited +7 oxidation state as existing in the form of ReO$_4^-$[41]. This analysis indicated that the Re species remain unchanged during the ion-exchange process. In order to verify the binding behavior of ReO$_4^-$ with adsorbents, the N 1$s$ spectra of MW-Ptriaz@MOF$_C$ and MW-Ptriaz@MOF$_C$(Re) were analyzed. As illustrated in Fig. 5e, five distinct types of N-containing functional groups can be resolved, presenting the positively charged nitrogen atom (N4), the non-ionic nitrogen atom (N1) and the naked nitrogen in 1,2,4-triazolium ring (N2), respectively, followed by the N atoms in cyano group (-C≡N) and N atoms in porphyrin ring which were coordinated with copper ions (N-Cu)[42]. After the adsorption, the binding energies of N4, N1, N2 and -C≡N increased from 402.35 eV, 401.04 eV, 400.09 eV and 399.33 eV to 402.59 eV, 401.52 eV, 400.71 eV and 399.93 eV, respectively (Fig. 5f). However, the binding energies of N-Cu barely changed. This suggested that a strong electrostatic interaction occurred between the cationic N-containing polymeric skeleton and negatively charged ReO$_4^-$. To further specify the interaction between cationic polymers and anions, $^1$H NMR measurements were employed. Figure 5g displayed the $^1$H NMR spectra of Ptriaz and the Ptriaz@4eq anions in a H$_2$O/DMSO-d$_6$ mixture (v/v = 1:5) solvent. The proton signal of C5 in the 1,2,4-triazolium ring was clearly observed at 10.3 ppm for Ptriaz. Upon the introduction of ReO$_4^-$ (molar ratio of Ptriaz/ReO$_4^-$ = 1:4), the C5-proton signal in the 1,2,4-triazolium ring shifted to high field at 10.18 ppm,

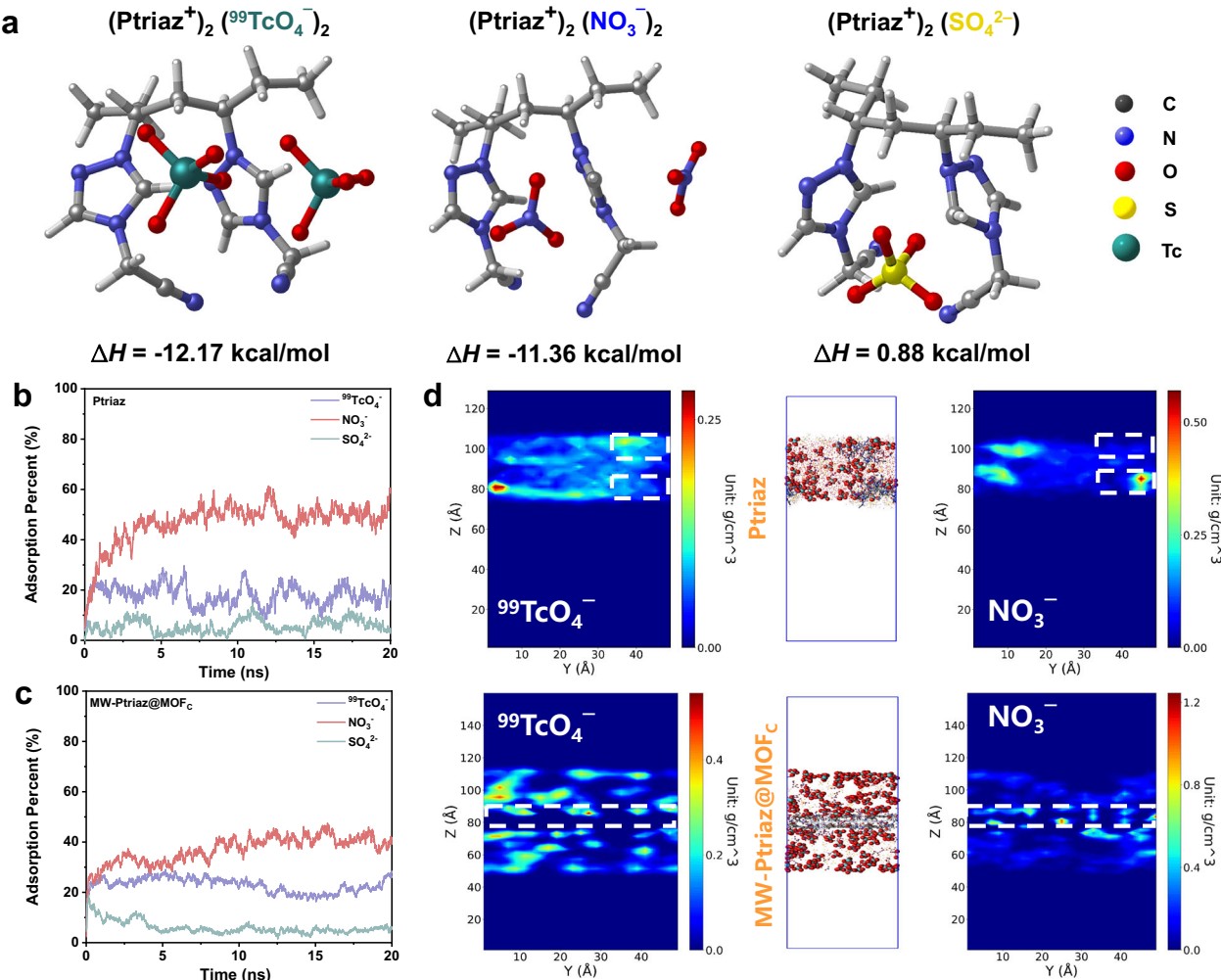

**Fig. 6 | Computational simulations of the adsorption of different anions into Ptriaz and MW-Ptriaz@MOF$_C$, respectively. a** Optimized models for $(Ptriaz^+)_2(^{99}TcO_4^-)_2$, $(Ptriaz^+)_2(NO_3^-)_2$, $(Ptriaz^+)_2(SO_4^{2-})$ and their corresponding enthalpy change values (ΔH). **b** Evolution of the adsorption ratio of Ptriaz over time (20 ns). **c** Evolution of the adsorption ratio of MW-Ptriaz@MOF$_C$ over time (20 ns). **d** Simulation snapshots of Ptriaz and MW-Ptriaz@MOF$_C$, and relevant two-dimensional mass density distribution of $NO_3^-$ and $^{99}TcO_4^-$ in the system after 20 ns, respectively. Source data are provided as a Source Data file.

which indicates the dynamic Lewis acid-base pair interaction of H···$ReO_4^-$ and electron shielding effect caused by enriched electron density of H···Re. However, on the contrary, the introduction of interfering anions leads to the shift of C5 proton towards low field (higher ppm value) in the $^1H$ NMR spectra. In comparison to $ReO_4^-$, the deshielding effect is dominant for other interfering anions with Ptriaz, in which the electron densities of the C5 protons tend to be reduced, presenting the weaker interactions between C5 protons and interfering anions. These results indicated that, other than electrostatic interactions, the enhanced electron density of H···Re leads to a stronger interaction between Ptriaz and $ReO_4^-$ revealing the superior adsorption capacity of Ptriaz towards $ReO_4^-$ and high selectivity of $ReO_4^-$ to interfering anions.

**Computational simulations of $^{99}TcO_4^-$ adsorption**

To reveal the adsorptive mechanism of $^{99}TcO_4^-/ReO_4^-$ in a theoretical manner, DFT calculations were performed to simulate the binding interactions between MW-Ptriaz@MOF$_C$ and $^{99}TcO_4^-/ReO_4^-$. According to aforementioned XPS analysis, the adsorption of $ReO_4^-$ by MW-Ptriaz@MOF$_C$ mainly depends on the three factors: electrostatic interactions between the cationic N-heterocycle rings and $ReO_4^-$, the affinity between cyano groups and $ReO_4^-$, together with the binding of

H···Re. As a result, we firstly investigated the adsorptive interactions between cationic polymer chains and relevant anions. The fragment of cationic polymer chains which contain the foremost structural features was selected as the theoretical models (abbreviated as $(Ptriaz^+)_2$). Electrostatic potential (ESP) analysis showed that the maximum positive potential appeared near the 1,2,4-triazolium rings in $(Ptriaz^+)_2$ (Fig. S48). In the authentic adsorption experiments, MW-Ptriaz@MOF$_C$ exhibited excellent adsorption selectivity for $ReO_4^-$ under the coexistence of $NO_3^-$ or $SO_4^{2-}$. Considering that, to further reveal the basis for this excellent selectivity, the optimized structures of $(Ptriaz^+)_2(^{99}TcO_4^-)_2$, $(Ptriaz^+)_2(NO_3^-)_2$ and $(Ptriaz^+)_2(SO_4^{2-})$ and their corresponding enthalpy change values (ΔH) were explored and applied to evaluate the affinity between anions and adsorbent[26]. As shown in Fig. 6a, all negatively charged anions were adsorbed on the positively charged 1,2,4-triazolium ring. Despite showing quite similar sorption sites on Ptriaz$^+$, their ΔH values were distinctly different. In detail, the calculated ΔH values for $(Ptriaz^+)_2(^{99}TcO_4^-)_2$ is −12.17 kcal mol$^{-1}$, which is higher than −11.36 kcal mol$^{-1}$ for $(Ptriaz^+)_2(NO_3^-)_2$ and 0.88 kcal mol$^{-1}$ for $(Ptriaz^+)_2(SO_4^{2-})$. These results revealed that $^{99}TcO_4^-$ is more favorable to be captured by positively charged Ptriaz$^+$, which can be corresponded to the more hydrophobic nature of $^{99}TcO_4^-$ instead of $NO_3^-$ and/or $SO_4^{2-}$.

Based on the preceding analyses, the confinement of cationic polymer chains within MOF nanochannels was expected to significantly enhance the adsorption efficiency of $^{99}TcO_4^-$. So, we next explored the selective adsorption behavior of $^{99}TcO_4^-$ against various competing anions (including $NO_3^-$ and $SO_4^{2-}$) on MW-Ptriaz@MOF$_C$ and Ptriaz in aqueous solutions by MD simulations. As shown in Fig. 6b, in Ptriaz, the adsorption ratio curves showed that, after 20 ns, the adsorption percent of $^{99}TcO_4^-$ kept consistent and stabilized at ~25%, and the adsorption ratio of $NO_3^-$ remained stable at ~55%. In contrast, for MW-Ptriaz@MOF$_C$, whose cationic polymer chains were confined within MOF nanochannels in an orderly manner, the adsorption percent of $^{99}TcO_4^-$ maintained essentially the same as that of the Ptriaz, while its adsorption ratio of competing anions ($NO_3^-$) reduced to ~40% (Fig. 6c). Impressively, it is also observed that very little $SO_4^{2-}$ (~5%) was adsorbed both on Ptriaz and MW-Ptriaz@MOF$_C$, which may also be accounted for the unfavorable enthalpy change values ($\Delta H = 0.88$ kcal/mol) of $(Ptriaz^+)_2(SO_4^{2-})$. In order to further elucidate the repulsive effect of MW-Ptriaz@MOF$_C$ on competing anions, the two dimensional mass density diagrams of anion distribution after adsorption were computed and visualized by Visual Molecular Dynamics over a predefined region in the simulation cell[43,44]. Figure 6d showed the simulation snapshots of Ptriaz and MW-Ptriaz@MOF$_C$ after immersing them in aqueous solution for 20 ns (water molecules not shown for clarity) and the corresponding 2D mass density distribution of $NO_3^-$ and $^{99}TcO_4^-$ on the YZ plane, respectively. The density heatmaps distinctly illustrated the relevant anion distribution of the system, with the deep blue color signifying the absence of the anions. In comparison to cationic polymers, an obvious decrease of $NO_3^-$ can be observed for the adsorption of MW-Ptriaz@MOF$_C$ in the simulation system, owing to the hydrophobic pore walls of MOF$_C$ who strongly repulse the highly solvated $NO_3^-$. These results suggested that the introduction of cationic polymer chains into the hydrophobic MOF nanochannels can strongly reduce the adsorption of competing anions, while still ensuring the adsorption efficiency of $^{99}TcO_4^-$.

## Discussion

In summary, we developed a general and mild synthetic strategy to construct hierarchically porous ionic polymer-MOF hybrid materials by confining and aligning flexible cationic polymer chains within MOF nanochannels through in situ molecular weaving. The synthesis of ionic polymer-MOF hybrid materials involves both shear forces generated by mechanical stirring and coordination bond-induced directional alignment. By employing dual-acting forces in molecular weaving, polymer chains are untangled by shear forces and realigned through directional coordination interactions. This process results in highly ordered cationic polymer chains confined within MOF nanochannels, as confirmed by physical characterization and theoretical calculations. Attributed to the well-tailored structure with dense concentration of anion-exchange sites, the as-synthesized molecularly woven ionic polymer-MOF hybrid materials are applied as adsorbents for radionuclide nuclear waste model anions ($ReO_4^-$), exhibiting excellent adsorption performance with high adsorption capacity, fast adsorption kinetics, high selectivity, stability, and reusability. In brief, a general approach for constructing ionic polymer-MOFs hybrid materials is reported, which offered us a new scavenger for the removal of radioactive anions from nuclear wastes and can be used as a versatile toolbox for synthesizing more hybrid porous materials in the future.

## Methods
### Materials and characterizations
All reagents and chemicals used in this study were obtained from Leyan Chemicals, Adamas-Beta Shanghai Titan Scientific Co., Ltd., and Sinopharm Chemical Reagent Co., Ltd. unless otherwise noted. Typically, 1-vinyl-1,2,4-triazole (98%) was purchased from Sigma-Aldrich. Furthermore, all chemical regents used in this work

were purchased from chemical reagent suppliers and used without further purification. For characterization, Fourier-transform infrared spectroscopy (FT-IR) spectra were collected using a Thermo Scientific Nicolet iS5 spectrometer. The scanning electron microscope (SEM) images were obtained on a Zeiss GeminiSEM 500. Transmission electron microscopy (TEM), annular dark-field STEM, and EDS elemental mapping were performed using a Talos F200S microscope. The specific surface areas of the samples were determined according to the BET model and the pore size distribution was calculated by the Non-Local Density Functional Theory (NLDFT) method. All relative data were obtained on a Micro ASAP 2046 surface area and porosity analyzer. XPS was conducted at a Thermo Fisher Scientific Escalab 250Xi under high vacuum ($1 \times 10^{-9}$ Torr), and all binding energy were calibrated to the C 1$s$ peak at 284.8 eV. Powder X-ray diffraction (PXRD) patterns were obtained on a Bruker D8 Advance diffractometer using Cu Kα radiation. The concentration of Re and other metal elements was measured on NexION 350D inductively coupled plasma mass spectrometer (ICP-OES PerkinElmer). Analytical DSC was performed on TA-DSC250 under a nitrogen atmosphere. Zeta potential data were obtained on SurPASS 3. The liquid proton nuclear magnetic resonance ($^1H$ NMR) measurements were taken at a Bruker AVANCE 3 HD 600 (600 MHz) spectrometer. The high-speed MAS $^1H$ solid-state NMR were taken at a Bruker AVANCE 400 (400 MHz) spectrometer.

### Synthesis of MW-Ptriaz@MOF$_A$
Ptriaz (0.241 mmol, 100 mg) and H$_3$BTC (1/3 eq, 0.0803 mmol, 16.87 mg) were dissolved in 1.5 mL DMF to form a homogeneous and transparent precursor solution (Solution-1). To prepare the complexation solvent (Solution-2), 1.5 mL triethylamine was added into 90 mL EtOH to adjust the pH to ~9–10. Solution-1 was then added dropwise into Solution-2 under vigorous stirring, leading to the immediate formation of insoluble complex precipitates. After 20 min, the mixture was ultrasonicated for 5 min and then stirred vigorously for another 5 min. The resulting precipitates were collected by filtration, washed with EtOH several times, and dried. For MW-Ptriaz@MOF$_A$ synthesis, the precipitates were added to a copper(II) chloride dihydrate solution (12 mL, DMF/EtOH/water = 1:1:1) with a Cu$^{2+}$/H$_3$BTC molar ratio of 3:1. The suspension was ultrasonicated for 1 min, then heated to 80 °C and stirred at 800 rpm for 6 h. The resulting hybrids were filtered, washed with EtOH and finally dried at 60 °C under vacuum.

### Synthesis of MW-Ptriaz@MOF$_B$
Ptriaz (0.241 mmol, 100 mg) and H$_2$BDC (1/2 eq, 0.1205 mmol, 20.0 mg) were dissolved in 1.5 mL DMF to form a homogeneous and transparent precursor solution (Solution-1). To prepare the complexation solvent (Solution-2), 1.5 mL triethylamine was added into 90 mL EtOH to adjust the pH to ~9–10. Solution-1 was then added dropwise into Solution-2 under vigorous stirring, leading to the immediate formation of insoluble complex precipitates. After 20 min, the mixture was ultrasonicated for 5 min and then stirred vigorously for another 5 min. The resulting precipitates were collected by filtration, washed with EtOH several times, and dried. For MW-Ptriaz@MOF$_B$ synthesis, the precipitates were added to a copper(II) chloride dihydrate solution (12 mL, DMF/EtOH/water = 1:1:1) with a Cu$^{2+}$/H$_2$BDC molar ratio of 2:1. The suspension was ultrasonicated for 1 min, then heated to 80 °C and stirred at 800 rpm for 6 h. The resulting hybrids were filtered, washed with EtOH, and finally dried at 60 °C under vacuum.

### Synthesis of MW-Ptriaz@MOF$_C$
Ptriaz (0.241 mmol, 100 mg) and TCPP (1/4 eq, 0.06025 mmol, 47.66 mg) were dissolved in 2.0 mL DMSO to form a homogeneous and

transparent precursor solution (Solution-1). To prepare the complexation solvent (Solution-2), 2.0 mL triethylamine was added into 90 mL EtOH to adjust the pH to ~9–10. Solution-1 was then added dropwise into Solution-2 under vigorous stirring, leading to the immediate formation of insoluble complex precipitates. After 20 min, the mixture was ultrasonicated for 5 min and then stirred vigorously for another 5 min. The resulting precipitates were collected by filtration, washed with EtOH several times, and dried. For MW-Ptriaz@MOF$_C$ synthesis, the precipitates were added to a copper(II) nitrate trihydrate solution (12 mL, DMF/EtOH = 3:1) with a $Cu^{2+}$/TCPP molar ratio of 3:1. The suspension was ultrasonicated for 1 min, then heated to 80 °C and stirred at 800 rpm for 4 h. The resulting hybrids were filtered, washed with EtOH, and finally dried at 60 °C under vacuum.

## Data availability

The authors declare that all the data supporting the findings of this study are available within the article and Supplementary Information. Source data are provided with this paper.

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

## Acknowledgements

This work was supported by the Fundamental Research Funds for the Central Universities (2232024Y-01), the National Natural Science Foundation of China (22102021, 22375037, 52073046, 52373172, 52103106), the National Key Research and Development Program of China (2023YFB3811100, 2022YFB3807100), the Program of Shanghai Academic Research Leader (21XD1420200), the Natural Science Foundation of Shanghai (23ZR1401100), the Key Technology Research and Development Program of Shanghai (25CL2900800) and the Chang Jiang Scholar Program (T2023082).

## Author contributions

W.-Y.Z. and Y.-Z.L. conceived and directed the research. X.-H.L. designed and carried out the experiments. X.L., F.C., Z.-Z.F., and Q.-H.H. contributed to the sample characterization. X.-H.L. and L.-L.Z. contributed to theoretical calculation. H.-W.W. and J.-Y.Y. joined the discussion of the data and gave helpful suggestions. X.-H.L. wrote the manuscript. Furthermore, all authors contributed to writing the paper and gave approval to the final version of the manuscript.

## Competing interests

The authors declare no competing interests.
