## [Transparent Peer Review file · Nature Communications]

In situ molecular weaving of ionic polymers into metal organic frameworks for radioactive anion capture

Corresponding Author: Professor Weiyi Zhang

Version 1:

Reviewer comments:

Reviewer #1

(Remarks to the Author)

The manuscript entitled " In situ molecular weaving of ionic polymers into MOFs via dual forces for efficient sequestration of radioactive anions " describes the synthesis of the ionic polymer-MOF hybrid materials by in situ molecular weaving and its uptake capability for ReO_4^- . The authors systematically investigated key parameters (pH, contact time, initial Re concentration, and competing anions) and demonstrated enhanced adsorption capacity of the hybrid material compared to pristine cationic polymers and MOFs. Although the work is interesting and scientifically sound, it is my opinion that the limited novelty does not render it suitable for Nature Communications. For example, polymer-MOF hybrid materials prepared by molecular weaving strategy have been published. What sets the strategy employed in this work apart in terms of uniqueness and innovation? Furthermore, the removal capability of ReO_4^- by this ionic polymer-MOF hybrid material lags considerably behind that of existing MOFs or polymers, particularly in terms of selectivity. Many sorbents could remove nearly 99% of Re on the experience of 100 times of NO_3^- , whereas the MW-Ptriaz@MOFC described herein achieves only 57 % removal of Re under the similar condition. Nevertheless, several issues are listed follow that I think the authors should address when considering publishing their work in the future.

1. Prior to the synthesis of the polymer-MOF hybrid materials, a complex of the deprotonated carboxylic coordination ligands and the cationic polymer chains was constructed. However, some related characterizations of this complex are missing in the existing manuscript.
2. The solid-state NMR peaks are not distinct, It is recommended to use peak-splitting software for data processing.
3. The author claimed that a significant decrease in the pore sizes was observed for MW-Ptriaz@MOFC compared to pristine and MOFC, However, Figure S12 reveals that MW-Ptriaz@MOFC exhibits pronounced hysteresis and larger pore sizes relative to the original MOFC material.
4. The SEM and SEM-EDS mapping of NS-Ptriaz@MOFB/C is suggested to be added in the revised manuscript.
5. In this manuscript, the synthesis and structure of MW-Ptriaz@MOFA are mainly studied, but MW-Ptriaz@MOFC is chosen for application analysis. Please explain the reasons.

Reviewer #2

(Remarks to the Author)

I have carefully read the manuscript "In situ molecular weaving of ionic polymers into MOFs via dual forces for efficient sequestration of radioactive anions". This manuscript is interesting and presents an innovative molecular weaving method for preparing polymer-MOF composite materials and discusses their potential applications in the treatment of radioactive waste liquid. I would recommend a decision of minor revision for this manuscript after the authors successfully address the following comments:

1. Molecular weaving technology, as the core innovation, needs to be more explicitly compared with existing synthesis strategies for MOF-polymer composites (such as in-situ polymerization, post-synthetic modification, etc.). For example, it is mentioned in the text that traditional methods require high-temperature activation and have low loading rates (page 3), but there is no quantitative comparison of the advantages of the method presented in this paper in terms of loading rate, energy consumption, or time cost.
2. Is it reasonable to use Re ions for analysis in adsorption experiments as a substitute for Tc ions? Are there potential

differences in the specific adsorption mechanisms and adsorption properties between the two, and are there differences in their practical applications?

3. Does the content of the polymer affect the adsorption performance of the overall composite material? It is currently unclear how much polymer is contained in the composite material, and no optimization of the polymer content has been conducted. Can molecular weaving achieve controllable adjustment of the polymer content?

4. The SI indicates that the material after adsorption is regenerated using NO_3^- solution, however, the authors state that the material still exhibits high selectivity and adsorption efficiency for ReO_4^- in NO_3^- solution. Is there a contradiction here? What is the specific regeneration mechanism?

5. The manuscript mentions that the hydrophobicity of the MOF contributes to the adsorption process, however, there is currently a lack of discussion on the specific contribution of MOF hydrophobicity to anion adsorption (for example, verifying the hydrophobicity of the MOF through contact angle tests).

Reviewer #3

(Remarks to the Author)

In this work, the authors reported a novel in situ molecular weaving strategy that utilizes two key forces (shear force and coordination interactions) to induce the controlled disentanglement and precise alignment of cationic polymer chains within the growing MOF nanochannels. The synthesized MW-Ptriaz@MOF hybrid materials demonstrate a uniform dispersion of cationic polymer chains and exhibit good sequestration performance for ReO_4^- . Overall, this approach presents a new pathway for the construction of ionic polymer-MOF hybrid materials. The study is interesting, and the conclusions are well-supported by the data. Therefore, I highly recommend the publication of this work in Nature Communications after addressing the minor revisions suggested below:

1. Compared to the in situ molecular weaving strategy, the authors mentioned that conventional methods for introducing small molecules during in situ polymerization often result in low polymer loading after the initial polymerization. Could the authors clarify and confirm this point?

2. To better highlight the in situ molecular weaving strategy driven by dual forces proposed in this work, it is suggested to move the conventional methods shown in Figure 1a to the Supplementary Information section.

3. Page 15 Line 318. The authors mentioned that upon desorption of ReO_4^- , the regenerated MW-Ptriaz@MOFC maintained good crystallinity, further corroborating its structural integrity. Could the authors provide information on whether there are any variations in the morphology and microstructure of the adsorbent after regeneration?

4. The authors mentioned that the Lewis acid-base pair interactions between the C5-H of Ptriaz and ReO_4^- were key to its high selectivity for ReO_4^- adsorption. Could the authors clarify how this interaction is proven to also be effective for the selective adsorption of $^{99}\text{TcO}_4^-$?

5. For the practical application of MW-Ptriaz@MOF, the column sorption investigation was encouraged to be performed.

Version 2:

Reviewer comments:

Reviewer #1

(Remarks to the Author)

Overall, the authors did a good job addressing most of the questions and suggestions by the reviewers, and as a result, the revised version of the manuscript is much improved. However, I believe further improvements could be realized with some minor revisions to the manuscript prior to publication.

1. The selective removal efficiency of MW-Ptriaz@MOFC demonstrated a marked improvement with an increase in the solid-to-liquid ratio. Nevertheless, this performance remains substantially inferior to most reported MOFs or polymers. Notably, numerous sorbents could remove nearly 99% of Re even in the presence of 100 times of NO_3^- or 1000 times of SO_4^{2-} . To highlight the advantages of the in situ molecular weaving strategy, it would be critical to provide comparative evidence showing that MW-Ptriaz@MOFC exhibits superior selectivity to the pristine Ptriaz polymer, MOFC, and MOF-polymer composites synthesized through existing methods.

2. The selectivity for multiple competing anions was evaluated using a mixed solution of 0.1 mM of ReO_4^- and 0.1 mM of each competing anion. However, the total amount of ReO_4^- and competing anions under this condition is still much lower than the uptake capacity of MW-Ptriaz@MOFC. Consequently, this experiment does not sufficiently demonstrate the material's high selectivity MW-Ptriaz@MOFC. Comparing the removal rates of MW-Ptriaz@MOFC for competing anion under this condition, or evaluating the adsorption capacity of MW-Ptriaz@MOFC for ReO_4^- at high concentrations, would offer a more accurate understanding of adsorption selectivity.

Reviewer #2

(Remarks to the Author)

The authors have thoroughly responded to all reviewers' concerns and provided additional data as needed. I think the manuscript is now acceptable for publication.

Reviewer #3

(Remarks to the Author)

The raised concerns have been well addressed. Please consider to accept the revised manuscript.

Version 3:

Reviewer comments:

Reviewer #1

(Remarks to the Author)

Authors have addressed my comments in full.

May 2025

Subject: Response Letter

Dear Reviewers,

Thank you very much for your valuable comments on our manuscript (NCOMMS-25-00071C) entitled "**In situ molecular weaving of ionic polymers into MOFs via dual forces for efficient sequestration of radioactive anions**". We greatly appreciate your critical suggestions. The manuscript has been carefully revised accordingly. Our point-by-point responses to the comments are listed below. The manuscript after this revision process, in our opinion, has significantly improved its quality and readability, and meets the requirement of reviewers. We hope that this revised manuscript would now be considered for publication in *Nature Communications* as a research article.

Your Sincerely,

Yaozu Liao & Weiyi Zhang

Comments from Reviewer #1

General comments: Overall, the authors did a good job addressing most of the questions and suggestions by the reviewers, and as a result, the revised version of the manuscript is much improved. However, I believe further improvements could be realized with some minor revisions to the manuscript prior to publication.

Response: We sincerely thank the Reviewer 1 for the positive and encouraging feedback. We have carefully addressed the remaining comments and made revisions to further improve the clarity and completeness of the manuscript. We hope that the current version meets the expectations for publication.

*Comment 1: The selective removal efficiency of MW-Ptriaz@MOFC demonstrated a marked improvement with an increase in the solid-to-liquid ratio. Nevertheless, this performance remains substantially inferior to most reported MOFs or polymers. Notably, numerous sorbents could remove nearly 99% of Re even in the presence of 100 times of NO_3^- or 1000 times of SO_4^{2-} . To highlight the advantages of the *in situ* molecular weaving strategy, it would be critical to provide comparative evidence showing that MW-Ptriaz@MOFC exhibits superior selectivity to the pristine Ptriaz polymer, MOFC, and MOF-polymer composites synthesized through existing methods.*

Response: We thank the Reviewer 1 for the constructive suggestion. To further highlight the advantage of our *in situ* molecular weaving strategy, we performed a systematic comparative adsorption study using MW-Ptriaz@MOFC, pristine MOFC, and polymer-MOF hybrids synthesized *via*

in situ polymerization and externally assisted infiltration. Regarding the pristine P triaz, we note that its counterion (TFSI⁻) readily undergoes ion exchange with more hydrophilic anions such as NO₃⁻ or SO₄²⁻. This exchange increases the solubility of P triaz in aqueous media, causing it to dissolve and thus making it unsuitable for solid-phase adsorption studies in competitive systems. In contrast, the *in situ* molecular weaving process confines P triaz chains within MOF nanochannels, effectively immobilizing the polymer chains and preserving its anion-exchange function under aqueous conditions. This structural confinement is critical to enabling the high selectivity observed with MW-P triaz@MOF_C. The experiments were conducted under adsorption conditions of 25 ppm ReO₄⁻ and a solid-to-liquid ratio of 1 g/L, with increasing concentrations of competing anions (NO₃⁻ or SO₄²⁻).

As shown in in **Figs. R1-R4**, MW-P triaz@MOF_C retained a high removal efficiency of 64.23% for ReO₄⁻ even in the presence of 500-fold excess NO₃⁻, whereas the corresponding removal efficiencies for pristine MOF_C, polymer–MOF hybrid synthesized *via in situ* polymerization, and polymer–MOF hybrid synthesized *via* externally assisted infiltration dropped to 0.04%, 3.06%, and 0.92%, respectively. A similar trend was observed with SO₄²⁻ as the competing anion, where MW-P triaz@MOF_C still achieved 48.32% removal efficiency at a SO₄²⁻/ ReO₄⁻ molar ratio of 500:1, compared to 0.03%, 0.56%, and 0.14% for the respective controls. These results confirm that the *in situ* molecular weaving approach in this work provides significantly enhanced ReO₄⁻ selectivity under strongly competitive conditions.

➤ *In situ* molecular weaving :

Fig. R1 Effect of excess competing anions on ReO_4^- removal performance of MW- $\text{P}(\text{triaz})@\text{MOF}_C$ synthesized *via in situ* molecular weaving. (a) ReO_4^- removal efficiency under increasing molar ratios of NO_3^- to ReO_4^- . (b) ReO_4^- removal efficiency under increasing molar ratios of SO_4^{2-} to ReO_4^- .

➤ *Pristine MOF_C*:

Fig. R2 Effect of excess competing anions on ReO_4^- removal performance of MOF_C . (a) ReO_4^- removal efficiency under increasing molar ratios of NO_3^- to ReO_4^- . (b) ReO_4^- removal efficiency under increasing molar ratios of SO_4^{2-} to ReO_4^- .

➤ ***In situ* polymerization:**

Fig. R3. Effect of excess competing anions on ReO_4^- removal performance of polymer–MOF hybrid synthesized *via in situ* polymerization. (a) ReO_4^- removal efficiency under increasing molar ratios of NO_3^- to ReO_4^- . (b) ReO_4^- removal efficiency under increasing molar ratios of SO_4^{2-} to ReO_4^- .

➤ ***Externally assisted infiltration:***

Fig. R4 Effect of excess competing anions on ReO_4^- removal performance of polymer–MOF hybrid synthesized *via* externally assisted infiltration. (a) ReO_4^- removal efficiency under increasing molar ratios of NO_3^- to ReO_4^- . (b) ReO_4^- removal efficiency under increasing molar ratios of SO_4^{2-} to ReO_4^- .

Comment 2: The selectivity for multiple competing anions was evaluated using a mixed solution of 0.1 mM of ReO_4^- and 0.1 mM of each competing anion. However, the total amount of ReO_4^- and competing anions under this condition is still much lower than the uptake capacity of MW-Ptriaz@MOFC. Consequently, this experiment does not sufficiently demonstrate the material's high selectivity MW-Ptriaz@MOFC. Comparing the removal rates of MW-Ptriaz@MOFC for competing anion under this condition, or evaluating the adsorption capacity of MW-Ptriaz@MOFC for ReO_4^- at high concentrations, would offer a more accurate understanding of adsorption selectivity.

Response: We sincerely thank Reviewer 1 for pointing out the need to further strengthen our demonstration of selectivity under competitive conditions. In addition to the equimolar (0.1 mM) mixed-anion experiments, we have previously investigated the adsorption performance of MW-Ptriaz@MOFC in a more complex and competitive environment—specifically, a simulated Hanford LAW melter recycle stream. This system represents a highly realistic waste matrix containing multiple competing anions at concentrations substantially higher than that of ReO_4^- , thereby offering a stringent test of the material's selective adsorption behavior.

As summarized in **Table R1**, the simulated stream contains NO_3^- , Cl^- , and NO_2^- at extremely high molar ratios relative to ReO_4^- (314:1, 330:1, and 873:1, respectively), far exceeding the conventional 100:1 benchmark typically used for evaluating competitive adsorption. **To further address the reviewer's concern**, we first performed additional single-ion competition experiments by systematically increasing the concentrations of NO_3^- , Cl^- , and NO_2^- over a wide range of molar ratios (1:1 to 500:1) with respect to ReO_4^- . As shown in **Figure R5-R7**, MW-Ptriaz@MOFC

consistently maintained high removal efficiencies (~90%) at a 100:1 molar ratio and retained appreciable performance (~60%) even at 500:1, highlighting its strong and selective affinity for ReO_4^- under intense ionic competition.

Furthermore, when exposed directly to the full simulated Hanford waste matrix, MW-Ptriaz@MOF_C still achieved a removal efficiency of 65.91% for ReO_4^- at a solid-to-liquid ratio of 10:1, underscoring its practical potential in highly competitive environments.

In general, we believe these results provide a more representative and demanding evaluation of selective adsorption and adequately address the reviewer's concern by demonstrating that MW-Ptriaz@MOF_C maintains strong selectivity for ReO_4^- even under realistic and highly complex conditions.

Table R1 Composition of simulated Hanford LAW Melter Recycle stream

Anion	Concentration, mol/L	Anion: $^{99}\text{TcO}_4^-/\text{ReO}_4^-$ molar ratio
$^{99}\text{TcO}_4^-/\text{ReO}_4^-$	1.94×10^{-4}	1.0
NO_3^-	6.07×10^{-2}	314
Cl^-	6.39×10^{-2}	330
NO_2^-	1.69×10^{-1}	873
SO_4^{2-}	6.64×10^{-6}	0.0343
CO_3^{2-}	4.30×10^{-5}	0.222

Fig. R5 Effect of excess competing NO_3^- anions on the removal efficiency of ReO_4^- .

Fig. R6 Effect of excess competing Cl^- anions on the removal efficiency of ReO_4^- .

Fig. R7 Effect of excess competing NO_2^- anions on the removal efficiency of ReO_4^- .

Comments from Reviewer #2

General comments: The authors have thoroughly responded to all reviewers' concerns and provided additional data as needed. I think the manuscript is now acceptable for publication.

Response: We sincerely thank the Reviewer 2 for the positive comment and support for the acceptance of our revised manuscript.

Comments from Reviewer #3

General comments: The raised concerns have been well addressed. Please consider to accept the revised manuscript.

Response: We sincerely thank the Reviewer 3 for the positive comment and support for the acceptance of our revised manuscript.